# An Artificial Intelligence Model for Sensing Affective Valence and Arousal from Facial Images

**DOI:** 10.3390/s25041188

**Published:** 2025-02-15

**Authors:** Hiroki Nomiya, Koh Shimokawa, Shushi Namba, Masaki Osumi, Wataru Sato

**Affiliations:** 1Faculty of Information and Human Sciences, Kyoto Institute of Technology, Kyoto 606-8585, Japan; nomiya@kit.ac.jp; 2RIKEN, Psychological Process Research Team, Guardian Robot Project, Kyoto 619-0288, Japan; koh.shimokawa@riken.jp; 3Department of Psychology, Hiroshima University, Hiroshima 739-8524, Japan; nashushi@hiroshima-u.ac.jp; 4KOHINATA Limited Liability Company, Osaka 556-0020, Japan; osumi@kohinet.com

**Keywords:** dimensional emotion rating, facial action units, machine learning, valence/arousal

## Abstract

Artificial intelligence (AI) models can sense subjective affective states from facial images. Although recent psychological studies have indicated that dimensional affective states of valence and arousal are systematically associated with facial expressions, no AI models have been developed to estimate these affective states from facial images based on empirical data. We developed a recurrent neural network-based AI model to estimate subjective valence and arousal states from facial images. We trained our model using a database containing participant valence/arousal states and facial images. Leave-one-out cross-validation supported the validity of the model for predicting subjective valence and arousal states. We further validated the effectiveness of the model by analyzing a dataset containing participant valence/arousal ratings and facial videos. The model predicted second-by-second valence and arousal states, with prediction performance comparable to that of FaceReader, a commercial AI model that estimates dimensional affective states based on a different approach. We constructed a graphical user interface to show real-time affective valence and arousal states by analyzing facial video data. Our model is the first distributable AI model for sensing affective valence and arousal from facial images/videos to be developed based on an empirical database; we anticipate that it will have many practical uses, such as in mental health monitoring and marketing research.

## 1. Introduction

Sensing subjective emotions using objective signals is useful in several ways. For example, emotional experiences are linked to mental and physical well-being [1] and business success, such as in product development [2]. Because subjective reports of emotional experiences can be biased and difficult to assess while performing active tasks, sensing emotions using objective signals can have practical applications [3].

Recent psychological studies have reported that dimensions of subjective emotions, such as valence and arousal, correspond to facial expressions [4,5]. Namba et al. [4] asked participants to remember various affective valence and arousal states and express them through facial expressions; they acquired images of participants’ faces and analyzed facial action units (AUs) using the automated version of the Facial Action Coding System (FACS) [6] and found associations between several AUs and valence or arousal states. Zhang et al. [5] videotaped participants’ faces and assessed dynamic valence and arousal ratings while they observed emotion-inducing films. Automated AU analysis revealed several associations between AUs and dynamic ratings of valence or arousal. Although associations between affective states and facial expressions have long been investigated based on categorical emotion theory [7], empirical research has not clearly supported the theory of facial expressions associated with emotional categories (for reviews, see [8,9,10]). These findings suggest that it is possible to sense affective dimensions from facial expressions.

Despite these findings, no artificial intelligence (AI) system has been developed to estimate affective dimensions based on firm evidence. Although many AI systems have been developed for sensing emotions from facial expressions, most of these are based on emotion categories [11,12]. Although the commercial AI system FaceReader can estimate dimensional affective states, it estimates these dimensional states indirectly, by learning the correspondence between facial expressions and emotion categories [13]. These AI systems have not learned to associate a person’s facial expressions with dimensional subjective affective states.

In this study, we developed an AI model that learns the correspondence between a person’s facial expressions and their subjective emotions. Our model receives the intensity values of several AUs as input and outputs affective dimension values (valence and arousal intensity) in real time. We used the RIKEN facial expression database [4] for training. We tested our AI model using facial video data provided by Zhang et al. [5]. We evaluated the performance of our model by conducting leave-one-out cross-validation and analyzing the correlation coefficients between actual and estimated valence and arousal values. We also compared the estimation performance of our model and FaceReader, a commercial software that outputs indirect affective dimension values. In addition, feature importance analyses were conducted using the drop-column importance method to interpret our model. Based on the AI model, we developed a graphical user interface (GUI)-based software to estimate affective dimension values. The application effectively visualizes the estimated intensity of valence and arousal.

## 2. Model Development

### 2.1. Model

An AI model was developed to estimate valence and arousal intensity using the intensity values of several AUs input into the model. AUs are the units of specific movements of facial muscles. As valence and arousal are thought to be closely related to the movement of facial muscles, we used AU intensity as the AI model input. Considering temporal changes in AU intensity, the input of the AI model was time-series data.

The AI model was constructed using a gated recurrent unit (GRU), such that time-series data could be used to train the model. A GRU is a type of neural network architecture used in machine learning, particularly for processing sequential data, like time series or natural language. It is a variant of recurrent neural networks (RNNs) and is designed to solve some of the challenges RNNs face, like the vanishing gradient problem, which makes it difficult for them to learn long-term dependencies in data.

Long Short-Term Memory (LSTM) is also widely used to train AI models with time-series data. However, GRUs have fewer parameters than LSTMs, making them faster and less resource-intensive. While simpler, GRUs are still capable of capturing long-term dependencies in data, similar to LSTMs. This simplicity often leads to faster training times and may reduce the risk of overfitting, especially with smaller datasets. In fact, we found through a preliminary experiment that the GRU was comparable to LSTM in estimation accuracy, and made slightly faster estimates than LSTM. Therefore, we adopted a GRU for the AI model. In another preliminary experiment, we found that training separate models for estimating valence and arousal was more effective in terms of estimation accuracy. Therefore, our AI model consists of separate GRU-based valence and arousal models.

Figure 1 shows the structure of the GRU-based model, which was used for both the valence and arousal models. To consider the temporal change in AUs, time-series AU intensity data were used as the model input. We used the 17 AUs shown in Table 1. The intensity values of these AUs were obtained using OpenFace. The time-series data were generated from five frames extracted from a 1 s facial video. Therefore, our model estimates valence and arousal intensity once per second. Using five frames appeared to be sufficient in the preliminary experiment.

### 2.2. Dataset

We used the RIKEN facial expression database [4], which contains facial videos of 48 Japanese participants, to train the AI model. In this database, the participants were recorded using 10 Kinect devices to capture multi-angle facial expressions, including both color images and 3D data. However, for the purpose of training, we adopted only the frontal color image data, considering the practical constraints of implementing an emotion estimation application under commonly available conditions.

In this database, the participants rated valence and arousal on a scale from 1 to 5, corresponding to the intensity of each emotion. For each participant, there are 25 videos corresponding to all combinations of the five levels of valence and arousal. Figure 2 shows the videos of a participant in this database. For the readability of the figure, this figure focuses on four patterns of extreme expressions, combining valence 1 or 5 with arousal 1 or 5, as well as one pattern of a neutral expression, resulting in a total of five expression patterns. Note that we used all 25 videos for each participant to train our model.

Each video consists of three phases: a 1-second transition from a neutral expression to the target expression, a 2-second maintenance phase of the target expression, and a 1-second return to the neutral expression. For each video, we extracted the 1-second segment in which the expression was most prominently displayed. Then, we used five frames extracted from the 1 s video at equal intervals. The extracted frames were converted into numerical data using OpenFace to obtain 17 action unit (AU) features, which were utilized as input for training. As the valence and arousal ratings were defined for each video, all five frames had the same rating.

### 2.3. Cross-Validation Performance

We evaluated the performance of the AI model using leave-one-out cross-validation based on the training dataset. In this experiment, facial videos of 1 participant were used for the test and those of the other 47 participants were used for training. The valence and arousal ratings were estimated for each facial video because such ratings were provided for each video in the training set. The optimal hyperparameters for the AI model were determined through preliminary analyses.

Figure 3 shows the correlation coefficients between actual and estimated valence and arousal values. The correlation coefficients were analyzed using single-sample *t*-tests against zero following Fisher transformation. The results showed that both valence and arousal rating prediction values were significantly positive ($t_{47}=9.88$ and 5.23, p<0.001, d=1.43 and 0.76).

Figure 4 and Figure 5 show the correlation coefficients between actual and estimated valence and arousal values for each participant, respectively.

The correlation coefficients of most of the participants are positive values. The correlation coefficients of almost 90% of the participants (43 participants) are positive for valence. As for arousal, the correlation coefficients of more than 80% of the participants (40 participants) are positive.

### 2.4. Feature Importance Analysis for Valence and Arousal Estimation

To evaluate the contribution of each AU to the prediction of valence and arousal, a feature importance analysis was conducted using the drop-column importance method. Drop-column importance quantifies the significance of individual features by removing them from the model and measuring the resulting change in prediction error. Specifically, if the removal of a particular feature leads to a significant increase in prediction error, it indicates that the feature plays a crucial role in the estimation task. Conversely, if the removal results in a decrease in error, it suggests that the feature may act as noise rather than contributing to the prediction.

For feature importance evaluation, the conventional absolute change in error was replaced with a relative fluctuation rate as an indicator. This approach involved normalizing the difference between the post-removal error and the original error using the original error value. By adopting this method, the influence of error magnitude was eliminated, allowing for a more accurate assessment of each feature’s relative contribution to the prediction task.

The drop-column importance of each AU is shown in Figure 6. The graph on the left side of the figure shows the drop-column importance for valence. In valence estimation, AU 4 (brow lowerer) and AU 1 (inner brow raiser) demonstrated particularly high importance. This finding aligns with previous studies that report a strong involvement of brow movements in the expression of valence [4], suggesting that the model effectively captures these established relationships. Additionally, AU 12 (lip corner puller) was also identified as a significant factor for valence estimation, reinforcing the well-documented role of smiling expressions as indicators of positive valence.

The graph on the right side of Figure 6 shows the drop-column importance for arousal. For arousal estimation, AU 4 (brow lowerer), AU 6 (cheek raiser), and AU 17 (chin raiser) exhibited high importance. This result suggests that in high-arousal states, increased facial muscle tension and activity, particularly in the cheeks and chin, contribute significantly to the model’s predictions. Furthermore, AU 7 (lid tightener) was identified as a highly important feature for arousal estimation, supporting the idea that expressions of surprise or tension are closely associated with high-arousal states.

The results obtained in this study correspond well with the insights discussed by Namba et al. [4] regarding the facial features associated with valence and arousal. This suggests that the features learned by the model align with actual emotional expressions, providing evidence that the proposed approach is a valid method for emotion estimation based on facial data.

The drop-column importance results are presented as numerical values for each AU. A positive value indicates that removing the feature increases prediction error, signifying that the feature is important for the model’s estimation task. In contrast, a negative value implies that removing the feature reduces error, suggesting that it may act as noise. Furthermore, larger absolute values indicate that the model is more dependent on that particular feature. By analyzing these feature importance values, this study provides insights into how the trained model determines valence and arousal. The identified important features align with findings from prior research, reinforcing the validity of the proposed emotion estimation approach.

## 3. Experiment

### 3.1. Dataset

We used the RIKEN facial expression database [4] to train our AI model, and facial videos of 23 Japanese participants from Zhang et al. [5] were used for the test. The facial videos from Zhang et al. [5] were recorded while the participants were watching five videos designed to evoke different emotions (anger, sadness, neutral, contentment, and amusement). The recordings were made using a single frontal camera, and the lengths of the videos were as follows: anger, 157 s; sadness, 172 s; neutral, 206 s; contentment, 148 s; amusement, 196 s.

In these videos, both segments with strongly expressed emotions and segments with weaker expressions were observed. After recording, the participants retrospectively evaluated their affective states during the recordings by subjectively rating valence and arousal using real numbers ranging from 1 to 9. For each facial video, second-by-second ratings were provided. Therefore, the valence and arousal ratings were estimated for each second. Since the valence and arousal ratings in the RIKEN facial expression database used for training ranged from 1 to 5, we adjusted the range of the output values to match the range from 1 to 9 used in the test dataset using the equation $I^{\prime }=2I-1$, where I(1≤I≤5) is the original rating and $I^{\prime }\;\left(1\leq I^{\prime }\leq 9\right)$ is the converted rating. This conversion ensured compatibility between the training and test datasets.

### 3.2. Experimental Settings

The AI model was implemented using Keras. The valence and arousal models were trained separately. We performed hyperparameter optimization for each of the valence and arousal models. The hyperparameters were optimized to minimize loss (mean absolute error of the correlation coefficients). The optimized hyperparameters are listed in Table 2; the default settings were used for all other hyperparameters.

We also compared the estimation performance of our model and FaceReader 9.0 (Noldus Information Technology, WCeningen, the Netherlands). The software detects the faces in the images based on the Viola–Jones algorithm [14], and constructs 3D face models based on the Active Appearance Method [15] in combination with deep artificial neural network classification [16]. Then, by using seven independent artificial neural networks trained on large databases of prototypical emotional facial expressions, the software quantifies the intensities of the seven emotion categories (i.e., six basic and neutral emotions) [17,18]. The software can also give the FACS scoring of a face [6] by using the similar artificial neural networks trained on large databases of AU-coded facial images [17,18]. Although the main output of the software is the intensity values of emotion categories, the software also calculates the valence and arousal values based on the category and AU values [13]. The valence is calculated as the intensity of ‘happy’ minus the intensity of the negative expression with the highest intensity [13]. For instance, if the intensity of ‘happy’ is 0.8 and the intensities of ‘sad’, ‘angry’, ‘fear’, and ‘disgusted’ are 0.2, 0.0, 0.3, and 0.2, respectively, then the valence is 0.8−0.3=0.5. The arousal is calculated based on the 20 AU values as follows [13]: (1) The 20 AU values are taken as input. These are AUs 1, 2, 4, 5, 6, 7, 9, 10, 12, 14, 15, 17, 20, 23, 24, 25, 26, 27, and the inverse of 43. (2) The average AU activation values are calculated over the last 60 s. During the first 60 s of the analysis, the average AAV is calculated over the analysis up to that moment. (3) The average AU activation values are subtracted from the current AU activation values as the corrected activation values. (4) The arousal is calculated from these corrected activation values by taking the mean of the five highest values.

### 3.3. Results

We estimated the valence and arousal ratings using the facial videos in the test dataset for every second. Then, the correlation coefficients between the actual and predicted valence and arousal ratings were computed for each participant using our model, as well as FaceReader 9.

Figure 7 shows the correlation coefficients between actual and estimated valence and arousal ratings. The coefficients were Fisher-transformed and the values were subjected to single-sample *t*-tests against zero. The results showed that the valence and arousal ratings of both our model and FaceReader were significantly positive ($t_{22}=7.01$, 5.12, 6.37, and 7.91, p<0.001, d=1.46, 1.07, 1.33, and 1.65, respectively).

Next, the z-transformed coefficients of our model and FaceReader were compared using paired *t*-tests. There were no significant differences in either valence and arousal ratings between the models ($t_{22}=0.80$ and 0.73, respectively, p>0.43). Bayesian paired *t*-tests confirmed the lack of significant differences (Bayes factor = 0.29 and 0.28, respectively).

The correlation coefficients of the 23 participants (participant 1 to 23) are shown in Figure 8 and Figure 9. Figure 8 shows the correlation coefficients for valence, and Figure 9 shows those for arousal.

As for the estimation of valence ratings, our model outperformed FaceReader in 8 participants out of 23 participants. FaceReader has higher correlation coefficients for most participants; there are a few participants whose correlation coefficients calculated by our model are much higher than those by FaceReader such as participants 17 and 20.

Regarding the estimation of arousal ratings, our model outperformed FaceReader in 16 participants out of 23 participants. In contrast to the estimation of valence ratings, our model shows better performance on average. However, the correlation coefficient of participant 9 is much lower compared to FaceReader. The test data by Zhang et al. [5] were created by showing five videos to the participants. The correlation coefficients are computed for the data including the five videos. When the correlation coefficients are separately computed for each video, the correlation coefficients of arousal for participant 9 are −0.113, 0.188, 0.004, 0.346, and 0.287. This stems from the high variance of the intensity values of arousal among the videos. This analysis result indicates that our AI model is comparable to FaceReader for every participant.

The feature importance analysis using drop-column importance was also conducted on the test data to identify the features contributing to the estimation of valence and arousal. The drop-column importance for each AU is shown in Figure 10. The graph on the left side shows the drop-column importance for valence and the graph on the right side shows that for arousal.

For valence estimation, as in the analysis for the training data, AU 2 (outer brow raiser), AU 1 (inner brow raiser), and AU 4 (brow lowerer) demonstrated particularly high importance. Additionally, AU 7 highly contributed to the valence estimation of the test set.

On the other hand, for arousal estimation, AU 4 (brow lowerer), AU 2 (outer brow raiser), AU 7 (lid tightener), and AU 1 (inner brow raiser) showed high importance. These features imply that increased facial tension in high-arousal states leads to more active movements of the brows and eyes. Additionally, AU 5 (upper lip raiser) and AU 6 (cheek raiser) also exhibited certain levels of importance, suggesting that muscle activity around the mouth may be involved in high-arousal states.

## 4. Application

We developed a GUI-based system to visualize the estimation results of our AI model. The system estimates and visualizes the intensity of valence and arousal in real time from a video of a single face (Figure 11). Either a video file or a video data captured by a web camera can be used for the facial video. The GUI-based system and the estimation model are separated, and we can choose an estimation model before the estimation. This design makes it easy to use an estimation model trained by a user.

The GUI system summarizes the estimation results as shown in Figure 12, including temporal changes in valence and arousal intensity throughout the entire video, and the distribution and frequency of intensity values, indicated by color changes. Our system is easy to use and will be helpful for the analysis of the estimation result of valence and arousal. Our system, named “KKR Facial Affect Reader”, is available for non-commercial academic purposes (https://github.com/prgshare/KKRFacialAffectReader (accessed on 12 February 2025)).

## 5. Discussion

Our cross-validation using the training dataset showed significantly positive correlations between actual and estimated valence and arousal intensity values. Validation based on the test dataset also showed significant positive correlations between actual and estimated valence and arousal ratings. In our analysis of the test set, the correlation coefficients of our AI model and FaceReader were comparable for both the valence and arousal ratings. These results suggest that our AI model can estimate affective dimensions (i.e., valence and arousal) comparably to the commercial AI system. However, our model learned the relationships between participants’ facial expressions and their subjective affective states, whereas FaceReader did not. To our knowledge, although there are a few AI models that can estimate dimensional affective states from facial images based on empirical data, ours is the first one that is publicly available.

Our AI model is anticipated to have practical value because emotional processing influences many aspects of daily life, including well-being and decision-making [19], and subjective ratings may be inappropriate or unavailable under some conditions [3], which would make estimating affective states using facial images valuable. For example, one clinical study reported that although patients with semantic dementia had difficulty reporting their affective states using rating scales while viewing emotional films, they produced evident facial muscle activity [20]. Because self-reports of emotional experiences could be biased, some marketing researchers recommended analyzing subtle facial muscle activity to predict customer behavior [21]. These findings suggest that our AI model will be useful for estimating emotional experiences in applied research.

One limitation of this study is that our analysis included only 17 AUs, whereas human FACS coders can evaluate more AUs [6]. Consequently, some associations between dimensional affective states and other AUs may have been missed. For example, Hyniewska et al. [22] investigated the link between emotional category recognition and AUs and identified a positive relationship between fear recognition and AU 16 (lower lip depressor), which was excluded from our analysis. Future AI models will likely analyze more AUs.

The results of the feature importance analysis using drop-column importance confirmed that specific AUs significantly contribute to the model’s predictions for valence and arousal estimation. For valence estimation, AU 2 (outer brow raiser), AU 1 (inner brow raiser), and AU 4 (brow lowerer) exhibited high importance. These features have also been reported in previous studies as crucial indicators in the perception of valence, suggesting that the model effectively learns the characteristics of human emotional expression.

Similarly, for arousal estimation, AU 4 (brow lowerer), AU 2 (outer brow raiser), and AU 1 (inner brow raiser) were found to be important, highlighting the strong influence of brow and eye tension on arousal perception. This finding reflects the fact that in high-arousal states, expressions of surprise and tension become more pronounced, making movements of the eyes and brows particularly noticeable. Additionally, AU 5 (upper lip raiser) and AU 6 (cheek raiser) also contributed to some extent, suggesting that muscle activity around the mouth may be involved in representing the intensity of emotions.

These results suggest that, as a characteristic specific to Japanese individuals, facial expressions may be more effectively conveyed through the eyes rather than the mouth. Additionally, while AU estimation using OpenFace has successfully captured AUs around the eyes, there are cases where it fails to accurately detect AUs around the mouth. Considering these findings, improving AU estimation not only for the eye region, which is already effective for emotion estimation, but also for the mouth region could lead to a more precise emotion recognition system. Further investigation into mouth movements is necessary to enhance the accuracy of emotion estimation.

Another limitation of this study is that we analyzed affective facial expressions while remembering personal events and watching film clips. Although these methods have advantages, such as allowing affect elicitation in laboratories, they lack the ecological validity of affective states. Further studies are needed to validate our model by analyzing other facial expression data recorded in realistic situations.

## 6. Conclusions

Despite the utility of sensing subjective emotions using objective signals, no AI model which is publicly available has been developed to estimate dimensional affective states from facial images based on empirical data. We developed a recurrent neural network-based AI model to estimate the affective dimensions valence and arousal. The effectiveness of our model was confirmed in a cross-validation experiment using the RIKEN facial expression database. We then trained our AI model using the RIKEN facial expression database and tested it with the facial datasets of Zhang et al. [5]. The results showed that our AI model demonstrated results comparable to those of FaceReader, a commercial software that depends on the estimation of emotion categories. This finding suggests that emotional categories and affective dimensions are not exclusively related. We also developed a GUI-based system to visualize estimated valence and arousal.

Although our AI model is comparable to FaceReader, further improvement of the estimation performance is essential because our model could not outperform FaceReader in the estimation of arousal. In addition, it was clarified through the experiment that the estimation performance of arousal as well as valence is not satisfactory considering the correlation coefficients of valence and arousal. Therefore, it is necessary to enhance the estimation performance of our model by improving the learning method. Another limitation of our AI model is that the number of AUs used for estimation was insufficient. Future research to develop a more effective AI model should analyze more AUs.

## Figures and Tables

**Figure 1 sensors-25-01188-f001:**
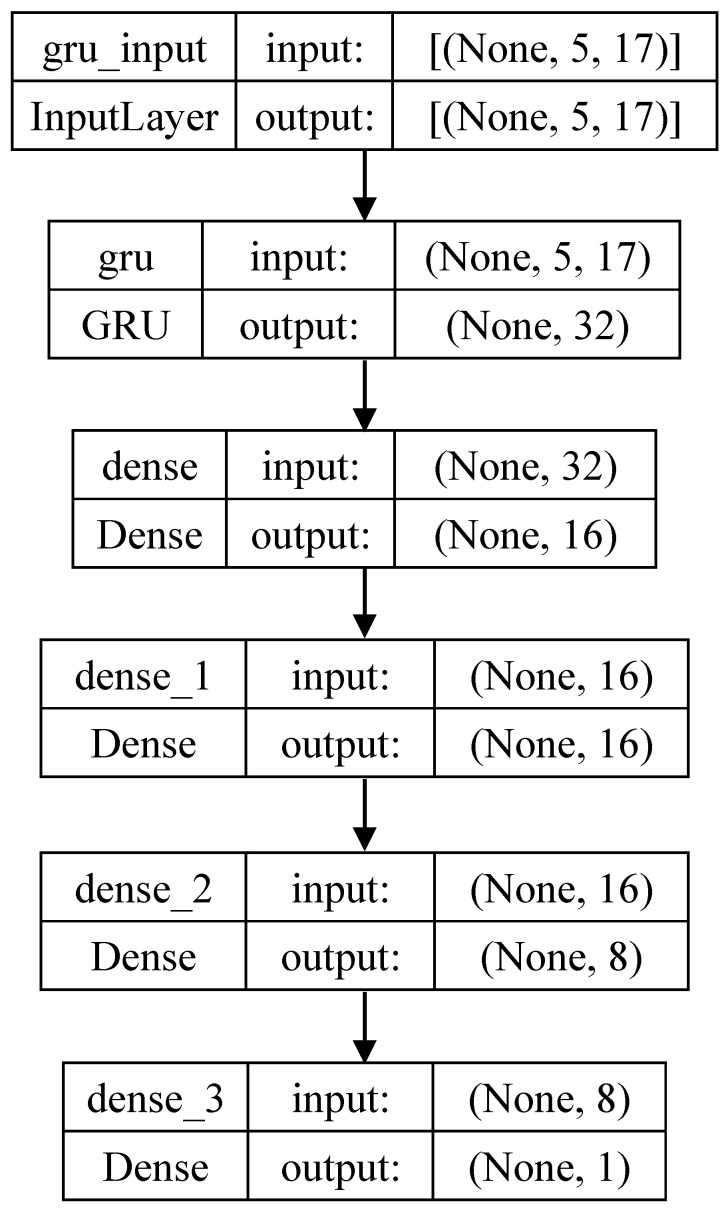
Structure of a gated recurrent unit (GRU)-based estimation model.

**Figure 2 sensors-25-01188-f002:**
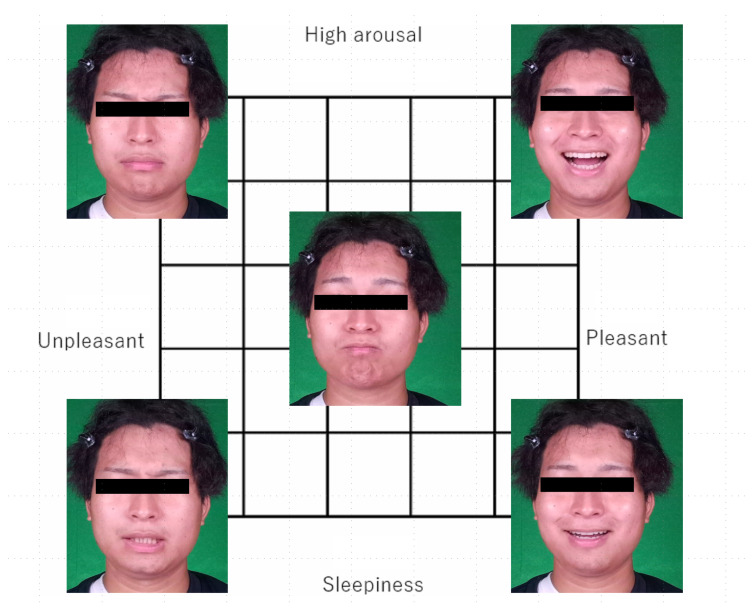
Videos of a participant in the RIKEN facial expression database.

**Figure 3 sensors-25-01188-f003:**
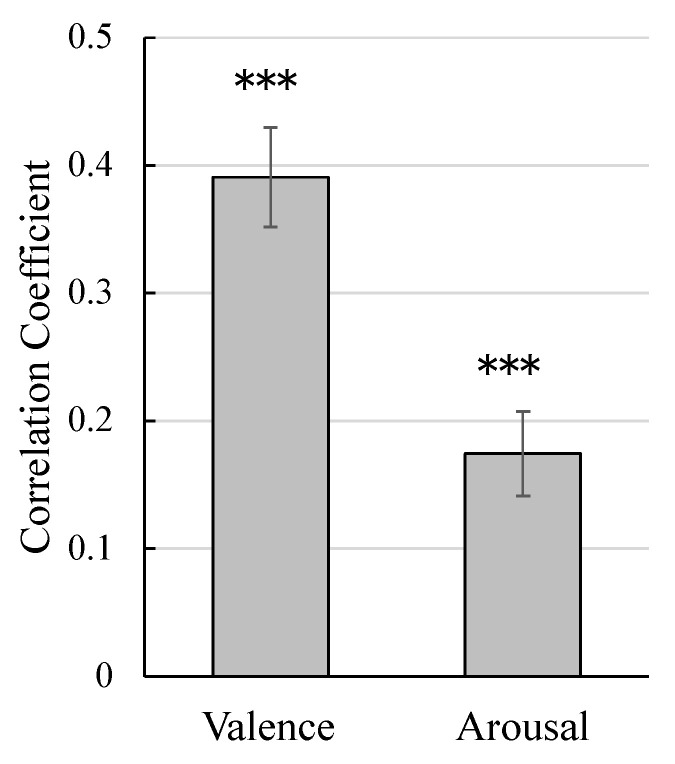
Mean (standard error) Pearson’s correlation coefficients between the actual and estimated ratings. *** p<0.001.

**Figure 4 sensors-25-01188-f004:**
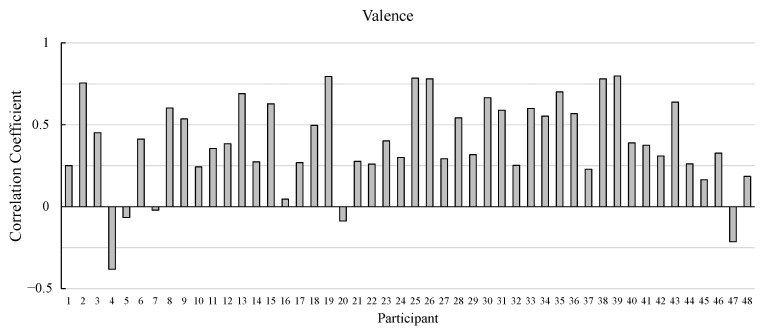
Pearson’s correlation coefficients between the actual and estimated ratings of valence in the model development.

**Figure 5 sensors-25-01188-f005:**
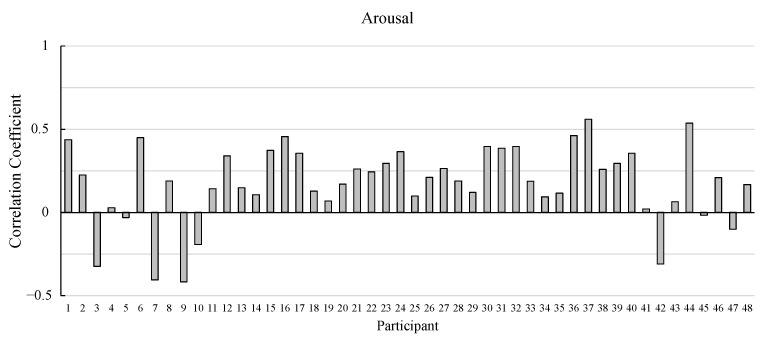
Pearson’s correlation coefficients between the actual and estimated ratings of arousal in the model development.

**Figure 6 sensors-25-01188-f006:**
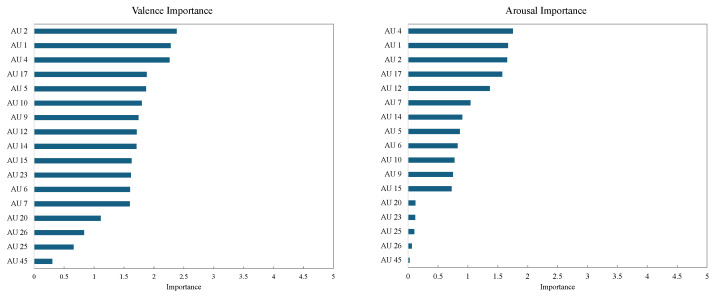
Results of drop-column importance for training data.

**Figure 7 sensors-25-01188-f007:**
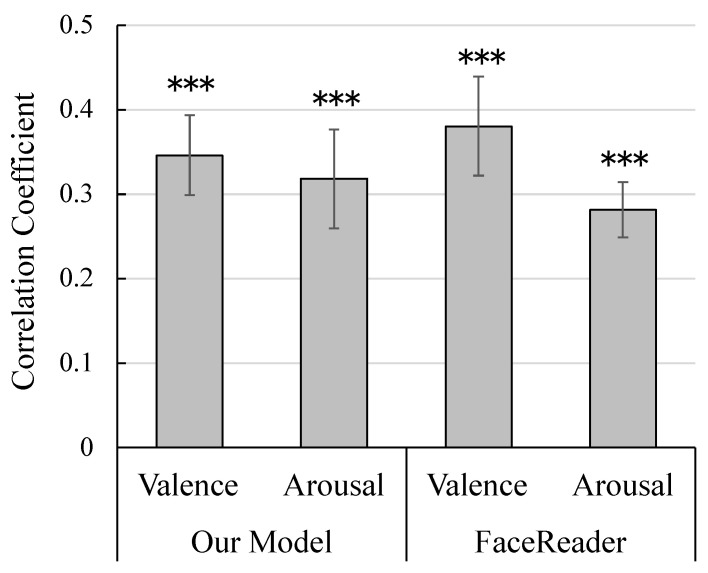
Mean (standard error) Pearson’s correlation coefficients between actual and estimated valence and arousal ratings estimated by our model and FaceReader 9. *** p<0.001.

**Figure 8 sensors-25-01188-f008:**
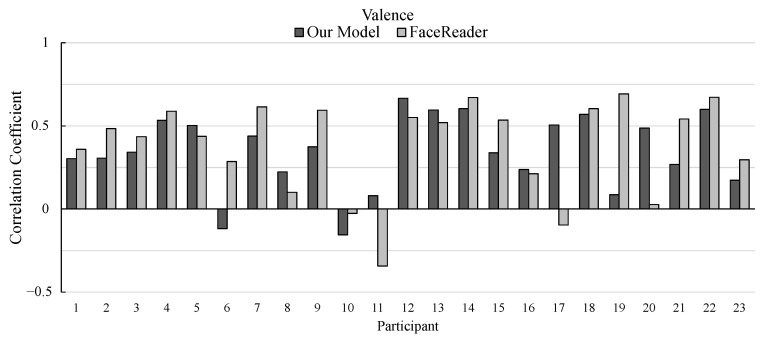
Pearson’s correlation coefficients between the actual and estimated ratings of valence in the experiment.

**Figure 9 sensors-25-01188-f009:**
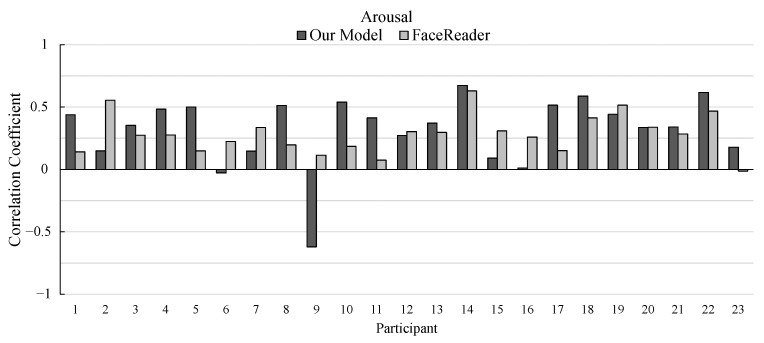
Pearson’s correlation coefficients between the actual and estimated ratings of arousal in the experiment.

**Figure 10 sensors-25-01188-f010:**
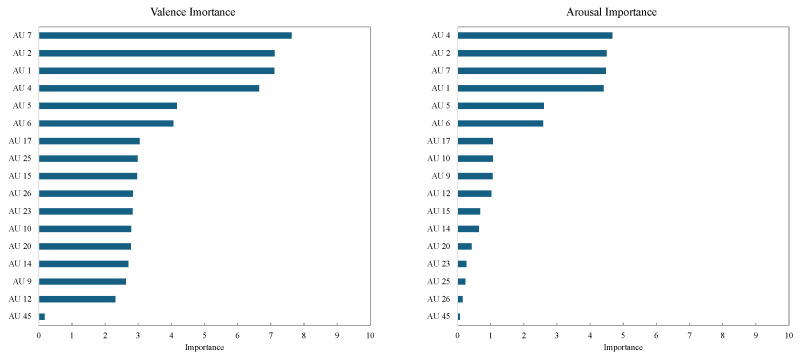
Results of drop-column importance for test data.

**Figure 11 sensors-25-01188-f011:**
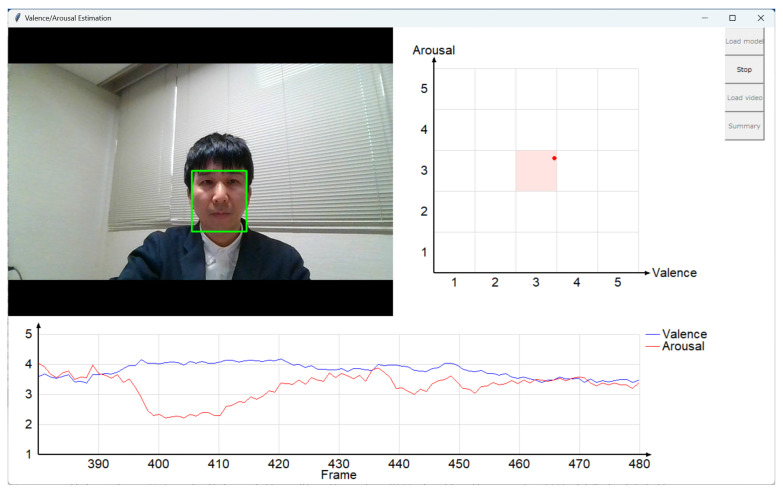
The GUI-based valence/arousal estimation system. **Top left**, input video; **top right**, current valence and arousal intensity values, represented as a point in two-dimensional space; **bottom**, graph showing changes in intensity values. The person shown is one of authors who agreed to show his face.

**Figure 12 sensors-25-01188-f012:**
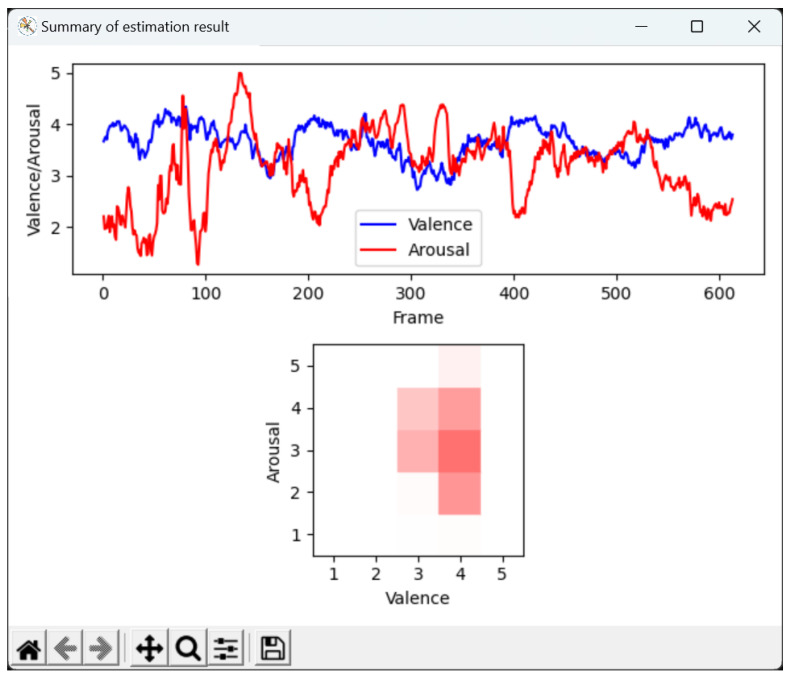
Summary of the estimation results. Top, temporal changes in valence and arousal intensity throughout the entire video. Bottom, distribution of intensity values. The two-dimensional space representing valence and arousal is divided into 5 × 5 regions; in each region, the frequency of the intensity value is represented by color intensity.

**Table 1 sensors-25-01188-t001:** Action units (AUs) used in the gated recurrent unit (GRU)-based estimation model.

AU Number	Description	Image
1	Inner Brow Raiser	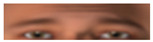
2	Outer Brow Raiser	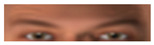
4	Brow Lowerer	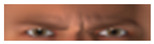
5	Upper Lid Raiser	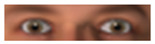
6	Cheek Raiser	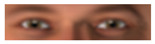
7	Lid Tightener	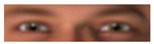
9	Nose Wrinkler	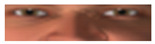
10	Upper Lip Raiser	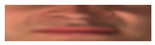
12	Lip Corner Puller	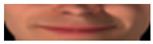
14	Dimpler	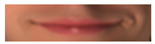
15	Lip Corner Depressor	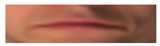
17	Chin Raiser	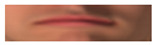
20	Lip Stretcher	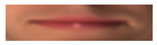
23	Lip Tightener	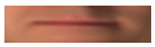
25	Lips Part	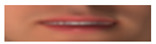
26	Jaw Drop	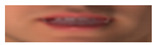
45	Blink	

**Table 2 sensors-25-01188-t002:** Hyperparameters of the valence and arousal models.

Hyperparameter	Valence Model	Arousal Model
activation	ReLU	None
dropout	0.530	0.680
recurrent_dropout	0.266	0.301
learning_rate	$1.23\times 10^{-5}$	$1.01\times 10^{-5}$
batch_size	48	32
epochs	100	100
loss	Mean absolute error	Mean absolute error

## Data Availability

The datasets analyzed during the current study are available from the corresponding author upon reasonable request.

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
