# Peer review of "An Artificial Intelligence Model for Sensing Affective Valence and Arousal from Facial Images"

_sensors, 2025, doi:10.3390/s25041188_

Round 1
Reviewer 1 Report
Comments and Suggestions for Authors
This paper is of particular interest as the authors present the development of a Recurrent Neural Network AI Model that can successfully detect emotions dimension valence and arousal. This model was evaluated for its effectiveness with the help of the RIKEN facial expression database with good results compared to other commercial software such as FaceReader, while the results depict that emotional categories and emotional dimensions are not exclusively related.
The presentation of the analysis of the data with bar graphs is unclear as regards the identification of values with respect to the vertical axis, where better grading is proposed.
This paper describes the comparison of a model proposed by the authors with the commercial application FaceReader. Although, based on the results, the proposed model is comparable to the commercial application, it appears that further improvement in the estimation performance of arousal is needed. The paper is extremely interesting and can be used as a basis for further research to improve the effectiveness of the proposed model.
Author Response
Dear Reviewer,
Thank you for your useful and constructive comments on our manuscript. We have carefully revised the manuscript according to your suggestions.
Point 1
The presentation of the analysis of the data with bar graphs is unclear as regards the identification of values with respect to the vertical axis, where better grading is proposed.
Response
In accordance with your suggestion, we have modified the wording of the bar chart. To make the meaning of the vertical axis easier to understand, we added the label “Correlation Coefficient” to the vertical axis. We have also added auxiliary lines to the graph to make it easier to read.
Sincerely yours,
Wataru Sato
Reviewer 2 Report
Comments and Suggestions for Authors
This paper provides a recurrent neural network-based AI model to estimate the emotion dimensions of valence and arousal.
The paper is well-organized and readable. I have some suggestions, described below, for improvement in the paper.
The authors used 17 Action Unit (AU) features representing eyes with eyebrows and lips. I would ask the authors to explain in more detail which emotion each of the AUs would belong to. The Action Unit does not tell us what our emotions are. It would be more interesting for the user to perceive the emotions of joy, laughter, sadness, or anger.
The authors have also prepared an application that they compare to the FaceReader application. The emotions of valence and arousal give me very little information about a person's well-being.
Are any 17 AU features compatible (for example, lips and eyes)? For example, do they express the same emotion? Did the authors combine any of these when defining emotions?
For better presentation and discussion of the results, I suggest that the authors should combine Figures 7 and 8 into one figure and Figures 9 and 10 into also in one figure.
I suggest you do some additional experiments on some more real-life emotions.
Author Response
Dear Reviewer,
Thank you for your useful and constructive comments on our manuscript. We have carefully revised the manuscript according to your suggestions.
Point 1
The authors used 17 Action Unit (AU) features representing eyes with eyebrows and lips. I would ask the authors to explain in more detail which emotion each of the AUs would belong to. The Action Unit does not tell us what our emotions are. It would be more interesting for the user to perceive the emotions of joy, laughter, sadness, or anger. The authors have also prepared an application that they compare to the FaceReader application. The emotions of valence and arousal give me very little information about a person's well-being. Are any 17 AU features compatible (for example, lips and eyes)? For example, do they express the same emotion? Did the authors combine any of these when defining emotions?
Response
According to your advice, to provide insight into how each AU could contribute to the estimation of Valence and Arousal, we have conducted Feature Importance analyses (pp. 5-7, 10-11). Specifically, Drop-Column Importance was employed to measure changes in prediction error when each AU was removed, thereby identifying which AUs contributed the most to the model's predictions.
Point 2
For better presentation and discussion of the results, I suggest that the authors should combine Figures 7 and 8 into one figure and Figures 9 and 10 into also in one figure.
Response
In accordance with your suggestion, we have combined Figures 7 and 8 into one figure. We have also combined Figures 9 and 10 into one figure.
Point 3
I suggest you do some additional experiments on some more real-life emotions.
Response
We agree that this study's lack of real-life emotions is a severe limitation. Due to the tight schedule for revising the manuscript, we could not conduct additional experiments. We have discussed this issue as a matter for future research.
Sincerely yours,
Wataru Sato
Round 2
Reviewer 2 Report
Comments and Suggestions for Authors
This paper provides a recurrent neural network-based AI model to estimate the affective dimensions of valence and arousal.
The paper is well-organized and readable. I have no more suggestions.
The authors considered some of my suggestions for improving the paper. The paper has been improved after the changes. Although the authors did not consider all my suggestions, I suggest the paper is suitable for publication.